# Non-neutralizing antibodies targeting the immunogenic regions of HIV-1 envelope reduce mucosal infection and virus burden in humanized mice

**Catarina E. Hioe** [1,2]*, **Guangming Li** [3,4], **Xiaomei Liu** [1], **Ourania Tsahouridis** [4], **Xiuting He** [3], **Masaya Funaki** [3], **Jéromine Klingler** [1,2], **Alex F. Tang** [1,5], **Roya Feyznezhad** [1], **Daniel W. Heindel** [1], **Xiao-Hong Wang** [6], **David A. Spencer** [7], **Guangnan Hu** [8], **Namita Satija** [1], **Jérémie Prévost** [9,10], **Andrés Finzi** [9,10], **Ann J. Hessell** [7], **Shixia Wang** [8], **Shan Lu** [8], **Benjamin K. Chen** [1], **Susan Zolla-Pazner** [1], **Chitra Upadhyay** [1], **Raymond Alvarez** [1], **Lishan Su** [3,4,11]

1 Division of Infectious Diseases, Department of Medicine, Icahn School of Medicine at Mount Sinai, New York, New York, United States of America, 2 James J. Peters VA Medical Center, Bronx, New York, New York, United States of America, 3 Laboratory of Viral Pathogenesis and Immunotherapy, Division of Virology, Pathogenesis, and Cancer, Institute of Human Virology, Department of Pharmacology, University of Maryland School of Medicine, Baltimore, Maryland, United States of America, 4 Lineberger Comprehensive Cancer Center, Department of Microbiology and Immunology, University of North Carolina, Chapel Hill, North Carolina, United States of America, 5 School of Medicine, University of California, San Francisco, California, United States of America, 6 VA New York Harbor Healthcare System–Manhattan, New York, New York, United States of America, 7 Division of Pathobiology & Immunology, Oregon Health & Science University, Oregon National Primate Research Center, Beaverton, Oregon, United States of America, 8 Department of Medicine, University of Massachusetts Medical School, Worcester, Massachusetts, United States of America, 9 Centre de recherche du Centre hospitalier de l'Université de Montréal (CRCHUM), Montreal, Quebec, Canada, 10 Département de Microbiologie, Infectiologie et Immunologie, Université de Montréal, Montreal, Quebec, Canada, 11 Laboratory of Viral Pathogenesis and Immunotherapy, Division of Virology, Pathogenesis and Cancer, Institute of Human Virology, Departments of Pharmacology and Microbiology & Immunology, University of Maryland School of Medicine, Baltimore, Maryland, United States of America

* catarina.hioe@mssm.edu, catarina.hioe@va.gov

**Data Availability Statement:** All relevant data are within the manuscript and its Supporting Information files.

## Abstract

Antibodies are principal immune components elicited by vaccines to induce protection from microbial pathogens. In the Thai RV144 HIV-1 vaccine trial, vaccine efficacy was 31% and the sole primary correlate of reduced risk was shown to be vigorous antibody response targeting the V1V2 region of HIV-1 envelope. Antibodies against V3 also were inversely correlated with infection risk in subsets of vaccinees. Antibodies recognizing these regions, however, do not exhibit potent neutralizing activity. Therefore, we examined the antiviral potential of poorly neutralizing monoclonal antibodies (mAbs) against immunodominant V1V2 and V3 sites by passive administration of human mAbs to humanized mice engrafted with CD34+ hematopoietic stem cells, followed by mucosal challenge with an HIV-1 infectious molecular clone expressing the envelope of a tier 2 resistant HIV-1 strain. Treatment with anti-V1V2 mAb 2158 or anti-V3 mAb 2219 did not prevent infection, but V3 mAb 2219 displayed a superior potency compared to V1V2 mAb 2158 in reducing virus burden. While these mAbs had no or weak neutralizing activity and elicited undetectable levels of antibody-dependent cellular cytotoxicity (ADCC), V3 mAb 2219 displayed a greater capacity to

**Funding:** This study was supported in part by NIH grant R01 AI139290 and VA Merit Review I01BX003860 to C.E.H, a CIHR foundation grant #352417 and NIH R01 AI148379 to A.F. C.E.H. is the recipient of the US Department of Veterans Affairs BLR&D Research Career Scientist Award IK6BX004607. A.F. is the recipient of a Canada Research Chair on Retroviral Entry #RCHS0235. J.P. is the recipient of a CIHR PhD fellowship. D.W.H. is supported in part by a Public Health Service Institutional Research Training Award NIH T32 AI07647. The funders had no role in study design, data collection and analysis, decision to publish, or preparation of the manuscript.

**Competing interests:** The authors have declared that no competing interests exist.

bind virus- and cell-associated HIV-1 envelope and to mediate antibody-dependent cellular phagocytosis (ADCP) and C1q complement binding as compared to V1V2 mAb 2158. Mutations in the Fc region of 2219 diminished these effector activities in vitro and lessened virus control in humanized mice. These results demonstrate the importance of Fc functions other than ADCC for antibodies without potent neutralizing activity.

## Author summary

In the past decade, HIV-1 has infected an estimated 1.5 to 2 million people every year, but vaccines needed to control this pandemic are unavailable. Among vaccines tested in the human efficacy trials, the RV144 vaccine regimen showed a modest efficacy and revealed non-neutralizing antibodies against the virus envelope glycoproteins as a correlate of reduced virus acquisition. To design more efficacious HIV-1 vaccines, a better understanding about antiviral mechanisms of these antibodies is needed. Here non-neutralizing monoclonal antibodies against two immunogenic sites on the virus envelope were evaluated for passive administration to humanized mice that were subsequently challenged with HIV-1. The antibodies did not block mucosal HIV-1 infection but reduced virus burden. The level of virus reduction correlated with the antibody binding potency and the effector functions mediated through their Fc fragments, which included antibody-dependent phagocytosis and complement activation, but not the commonly studied antibody-dependent cellular cytotoxicity. The importance of the Fc functions was further demonstrated by reduced virus control when mutations were introduced to decrease Fc activities. This study provides new evidence for the important contribution of multiple Fc-dependent antibody functions in immune control against HIV-1.

## Introduction

Almost forty years after the identification of HIV-1 as the virus that causes AIDS, ~38 million people worldwide are living with the virus [1]. Despite the achievement of effective virus suppression with combination antiretroviral therapies (cART) and improvements in prevention strategies that incorporate cART for treatment as prevention, pre-exposure prophylaxis, and post-exposure prophylaxis, 1.7 million new infections still occurred in 2019, disproportionately affecting populations with limited access to care. Preventive vaccines would be powerful tools for ending this epidemic, but none are yet available and the development of HIV-1 vaccines has faced tremendous scientific challenges. To generate efficacious vaccines, a better understanding is required of protective immune components and functions.

Of the phase IIb/III HIV-1 vaccine efficacy trials, the Thai RV144 (ALVAC/AIDSVAX) trial is the only one yielding a promising efficacy signal [2]. Although the vaccine efficacy of 31% was modest, this trial provided the first indication of vaccine-induced immune correlates for protection against HIV-1 in humans. Among the six primary variables measured, high IgG levels against the V1V2 region of HIV-1 Env was identified to be a correlate of reduced HIV-1 acquisition risk [3–5]. Subsequent studies defined additional correlates that include antibodies against the V3 loop in a subset of vaccine recipients with lower levels of Env-specific plasma IgA and neutralizing antibodies [6,7]. Nonetheless, the mechanistic correlates for protection remain unclear. The RV144 vaccine-induced antibody responses did not display broad or potent virus-neutralizing activity. Instead, neutralization of tier 1 HIV-1 strains and high levels

of antibody-dependent cellular cytotoxicity (ADCC) in combination with lower plasma anti-Env IgA were detected among the secondary correlates [3]. Comparison of RV144 (ALVAC/AIDVAX) with VAX003 (AIDVAX alone) trials subsequently revealed that anti-V1V2 IgG3 correlated with a reduced infection risk in the RV144 trial and that the IgG3 responses were associated with high ADCC activities [8]. Many Env-specific monoclonal antibodies (mAbs) isolated from the RV144 trial participants also display ADCC activity [9]. Additionally, antibody-dependent cellular phagocytosis (ADCP) linked to anti-V1V2 IgG3 and IgG1 was induced in RV144 but not in VAX003 [10]. In the HVTN 505 (DNA/rAd5) trial that lacked vaccine efficacy but showed immune pressure on the infecting viruses, a decreased HIV-1 risk correlated with ADCP, antibody binding to FcɣRIIa, and anti-Env IgG3 breadth [11]. A recent study comparing gene signatures induced in RV144 or RV306, both testing the same ALVAC/AIDSVAX regimen, with those in HVTN 505 (DNA/rAd5) also implicated ADCP by myeloid cells as a potential protective mechanism [12], further signifying the importance of non-neutralizing Fc-mediated antibody functions in vaccine-induced protection.

A previous study evaluating passive transfer of a non-neutralizing anti-gp41 mAb F240 in rhesus macaques demonstrated sterilizing protection in 2 of 5 animals and lower viremia in the remaining 3 animals after a high challenge dose of tier 2 SHIV SF162P3 [13]. In another non-human primate study, a combination of two non-neutralizing anti-gp41 mAbs formulated in a microbicide gel for topical vaginal administration showed no impact on SHIV SF163P3 acquisition upon vaginal challenge, but blunted peak viremia in 2 of 6 animals and caused a delay in one animal [14]. Yet, the passive administration of non-neutralizing anti-gp41 mAb 7B2 to rhesus macaques prior to SHIV BaL mucosal challenge showed no protection, although the number of transmitted/founder (T/F) variants was reduced [15]. The infusion of a non-neutralizing anti-gp41 mAb 246D to humanized mice with established HIV-1 infection also selected for escape mutation [16]. In contrast, sterile protection was achieved by passive transfer of a V3-specific mAb KD247 into cynomolgous macaques challenged with SHIV strain C2/1, which was neutralized by KD247 at an IC50 value of 0.5 μg/ml [17]. Administration of the V1V2-specific mAb 830A to rhesus macaques was also found to protect 5 of 18 animals that were repeatedly challenged with SHIV BaL and reduced plasma and cell-associated virus loads in blood and tissues of the remaining animals [18]. However, V1V2 mAb 830A neutralizes the tier 1 SHIV BAL challenge virus with an IC50 value of 1.4 μg/ml, so whether neutralizing and/or non-neutralizing activities mediated the reduced infection was not clear. Until now, no studies have evaluated the in vivo efficacy of V1V2- and V3-specific antibodies with no or poor neutralizing activity against tier 2 viruses which represent the majority of HIV-1 isolates. Therefore, in this study we sought to evaluate their protective potential against tier 2 HIV-1 by passive administration of non-neutralizing V1V2- and V3-specific mAbs to human CD34+ hematopoietic stem cell-engrafted mice capable of supporting HIV-1 infection.

In the present study, two human IgG1 mAbs, V1V2-specific 2158 and V3-specific 2219, were tested against an HIV-1 infectious molecular clone (IMC) with the tier 2 JRFL Env. MAb 2158 is specific for a conformation-dependent V2i epitope in the underbelly of the V1V2 domain near the integrin α4β7-binding motif [19–22]. MAb 2219 recognizes the crown of the V3 loop by a cradle-binding mode [23–25]. Both mAbs show a high degree of cross-reactivity with gp120 proteins from the major HIV-1 clades (A, B, C, D, F) and CRF02_AG, but the epitopes are masked in the functional Env spikes on most tier 2 HIV-1 virions, resulting in their inability to neutralize virus in the conventional in vitro assay [21,26–30]. Indeed, neutralization screening against large arrays of HIV-1 pseudoviruses with tier 1–3 Envs from different clades showed that 2158 neutralizes only a few tier 1 strains and 2219 neutralizes <50% of tier 1 and tier 2 strains [21,26,27], even though pseudoviruses are more sensitive to neutralization

than replication-competent viruses such as the JRFL IMC examined in this study. It is important to note, however, that unlike epitopes recognized by broadly neutralizing antibodies (bNAbs), the V2i and cradle V3 epitopes represent immunogenic sites that can be readily targeted by vaccination. This was evident from the detection of antibody responses against these specific epitopes in the vaccine recipients who participated in the VAX003, VAX004, and RV144 clinical trials, although durable responses remained unattainable [5,7,31–34].

In this study we examined plasma viremia and tissue-associated viral RNA (vRNA) and viral DNA (vDNA) burden in humanized mice that received V2i mAb 2158 or V3 mAb 2219 and were challenged with JRFL IMC via the rectal route. We determined the importance of Fc-mediated functions by administering V3 mAb 2219 with Fc mutations that significantly decrease Fc receptor and/or complement binding to humanized mice to protect against JRFL IMC. The data demonstrate the contribution of antibody-dependent cellular phagocytosis (ADCP) and complement-dependent functions to suppress infection of neutralization-resistant HIV-1, thus providing an impetus for the development of vaccine strategies that harness these Fc-dependent antibody functions to control HIV-1 infection.

## Results

### Passively administered V2i mAb 2158 and V3 mAb 2219 display differential activities against rectal HIV-1 challenge in CD34+ HSC-engrafted humanized mice

To assess the protective potential of anti-HIV-1 mAbs with poor or no neutralizing activities, we passively administered V2i mAb 2158 and V3 mAb 2219 to CD34+ HSC-engrafted humanized NSG mice that were then challenged intrarectally with a Δvpr HIV-1 IMC expressing the tier 2 JRFL envelope [35]. A third group of mice was given an irrelevant control mAb, 860, which is specific for the major capsid protein VP2 of parvovirus B19 [36]. Experiments were performed with three cohorts of mice generated with different HSC donors (**S1 Table**). Each mouse was given two doses of each mAb (700 μg/dose) intraperitoneally and challenged rectally with two doses of JRFL IMC at 700 $TCID_{50}$ (50% tissue culture infectious dose). The rectal challenge was chosen to represent a mucosal route pertinent for transmission to males and females. A total dose of 1400 $TCID_{50}$/animal was determined by titration in a prior experiment to yield 100% infection in all exposed animals within a week. The mAb half-life was also assessed in the plasma following transfer to uninfected mice and found to be 11 days for V3 mAb 2219 and >14 days for V2i mAb 2158 following a plateau at 35 and 30 μg/ml for the respective mAbs (**S1 Fig**).

In the first experiment, five mice per group were tested as outlined in **Fig 1A** and measurement of plasma vRNA loads showed that infection was established in all mice within 2 days of virus challenge (**Figs 1B and S2A**). The vRNA loads of mice receiving the control mAb 860 or V2i mAb 2158, were similarly high from day 2 to day 9, although a brief but significant drop in the average vRNA load was observed on day 4 in the V2i mAb 2158-treated animals compared to the mAb 860 control group (**Fig 1B**). In contrast, the recipients of V3 mAb 2219 displayed a significant reduction in vRNA loads at days 4, 7, and 9 as compared to the 860 control group, with three of the five 2219-treated animals having viral loads below detection (**S2A Fig**). To compare data from three cohorts of mice tested in separate experiments, areas under the curve (AUC) of the longitudinal vRNA loads were calculated relative to the mean AUC of the control 860 group. Although a high degree of variability was observed among individual animals, the data demonstrated the same overall pattern. The virus burden of the 2158-treated group was slightly and insignificantly lower than that of the 860 control group, whereas the recipients of V3 mAb 2219 had a significant reduction in vRNA loads (**Fig 1C**).

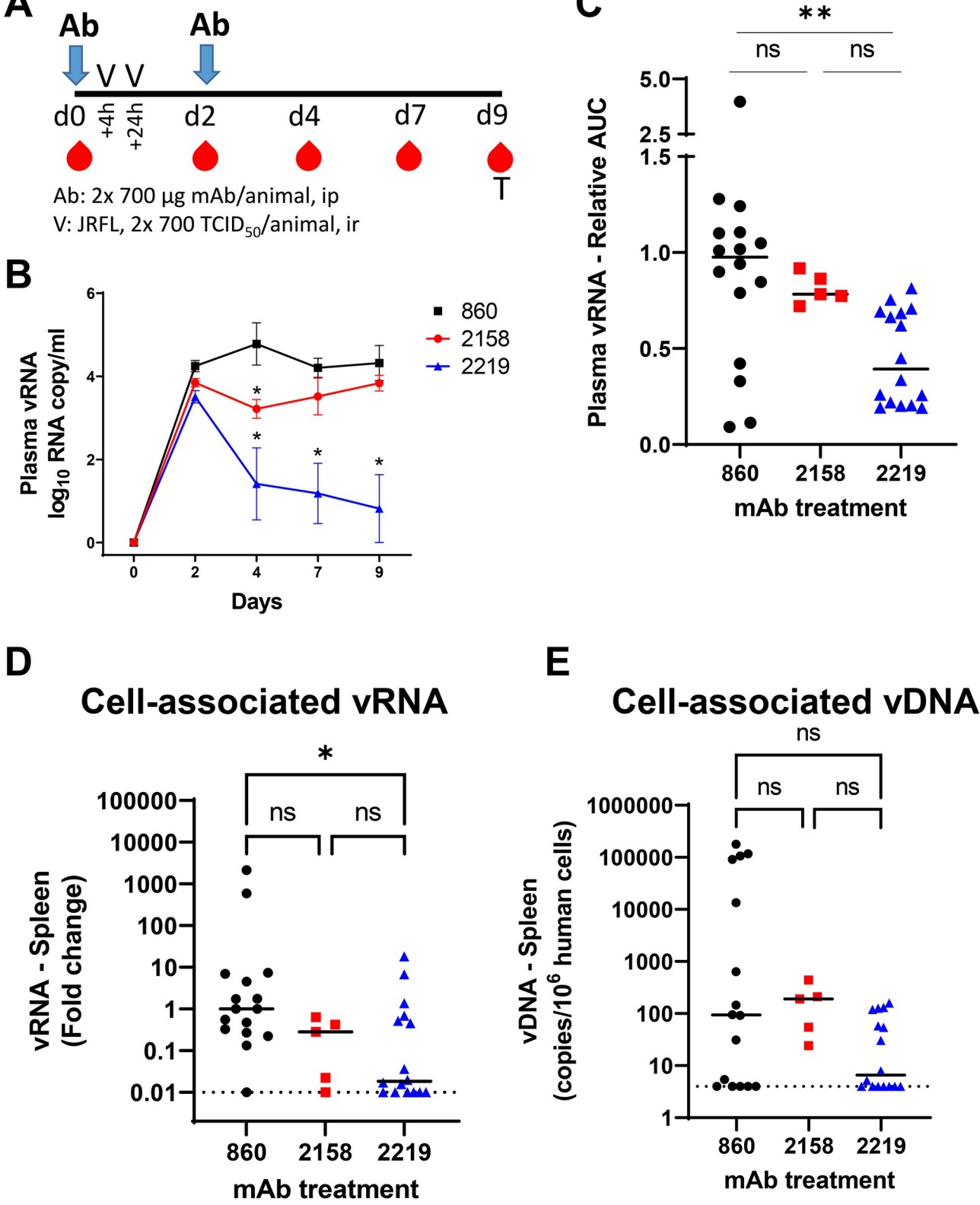

**Fig 1. The capacity of V2i mAb 2158 and V3 mAb 2219 to reduce virus infection in CD34+ HSC-engrafted humanized mice upon rectal challenge with HIV-1 JRFL IMC.** A) Schematic of experimental protocol showing that mice received intraperitoneal administration of mAb at day 0 and were challenged rectally with HIV at 4 hours and 24 hours. Animals received the second dose of mAb at day 2. Virus dose (2x700 TCID50 per animal) was

pre-determined to yield infection in all control mice. Blood and tissue samples were collected at the designated days for measurement of vRNA and vDNA in individual animals. B) Mean plasma vRNA loads from day 0 to day 9 in each group of animals that received control mAb 860, V2i mAb 2158, or V3 mAb 2219. Data from one of three experiments are shown. C) Plasma vRNA loads of individual mice in each of the three groups. Area under the curve (AUC) of vRNA over time was calculated. Data are presented as relative AUC over mean AUC of the control 860 group included in each experiment. Three experiments were performed with cohorts of mice engrafted with different HSC donors. N = 16, 5, and 16 for mice treated with 860, 2158, and 2219, respectively. D) Relative levels of cell-associated vRNA detected in the spleen collected at the end of experiment from individual mice in the three groups. Data from three experiments are compiled and presented as fold changes over median vRNA of the control 860 group in each experiment. E) Cell-associated vDNA levels in the spleen of mice from the three groups. Data are shown as vDNA copies per $10^6$ human CD45+ cells. Statistical analysis was done with Kruskal Wallis one-way ANOVA test with Dunn's multiple comparison. $^*$, p<0.05; $^{**}$, p<0.01; ns, not significant (p >0.05). Horizontal bars: median. Dotted lines: detection limit.

After termination of the experiment, the levels of cell-associated vRNA and vDNA in tissues were quantified. Overall the vRNA and vDNA levels in the spleen were highly variable among individual animals. However, the spleen vRNA amounts of the V3 mAb-treated 2219 group were significantly lower than those of the control 860 group (**Fig 1D**). Lower vDNA levels was also observed, but the decrease was not statistically significant (**Fig 1E**). In the V2i mAb 2158 group, however, we observed no significant reduction of either vRNA or vDNA in the spleen as compared to the 860 control group (**Fig 1D and 1E**). Measurement of vRNA and vDNA in bone marrow and mesenteric lymph nodes collected from one of the experiments showed similar results. As compared to the 2158-treated group, the 2219-treated mice more consistently exhibited reduction of vRNA and vDNA in these tissues (**S2B and S2C Fig**).

These data demonstrate that although the passive transfer of V3 mAb 2219 and V2i mAb 2158 failed to prevent the establishment of virus infection, these mAbs showed the capacity to decrease virus burden. The V3 mAb 2219 exhibited a superior potency to control HIV-1 infection in vivo compared to V2i mAb 2158, suggesting a differential capacity to mediate effector functions.

## V3 mAb 2219 binds better to virion- and cell-associated Env than V2i mAb 2158

Next, the V3 mAb 2219 and V2i mAb 2158 were examined for the ability to recognize different forms of HIV-1 Env. Both mAbs were isolated from US subjects infected with clade B viruses [26]. 2219 and 2158 displayed strong ELISA reactivity with recombinant soluble gp120 proteins of JRFL and several other strains from different HIV-1 clades (**Fig 2A and 2B**). The half maximal effective concentrations ($EC_{50}$) of 2219 were 1.0- to 2.9-fold lower than 2158, depending on the HIV-1 gp120 strains tested (**Fig 2B**). Kinetics analysis by biolayer interferometry further showed that while V3 mAb 2219 and V2i mAb 2158 have similar KD values in the picomolar range for recombinant JRFL gp120, V3 mAb 2219 has a 14-fold faster Kon rate compared to V2i mAb 2158 (**Fig 2C**). Interestingly, V3 mAb 2219 also demonstrated as much as 8-fold greater binding to virion-derived solubilized Env from JRFL IMC than V2i mAb 2158 (**Fig 2D and 2E**), although the relative binding to other solubilized Envs ranged from 5-fold weaker for REJO to 18-fold stronger for BG505 (**Fig 2E**). In these ELISA assays, recombinant gp120 proteins were directly coated onto ELISA plates (**Fig 2A and 2B**), whereas solubilized Env in virus lysate was captured by ConA coated on the plates (**Fig 2D and 2E**) before incubation with 2219 or 2158.

Having compared the binding of 2219 and 2158 to soluble Env, we investigated their abilities to bind Env on virion and cell surfaces. The differences between 2219 and 2158 were more evident in their ability to bind virion- and cell-associated Env (**Fig 3**). As compared to 2158, 2219 was observed to capture significantly higher levels of free JRFL virions (**Fig 3A and 3B**). Virus capture was measured after virus-mAb incubation for 24 hours. 2219 also bound cell surface-expressed JRFL Env (**Fig 3C**) and CD4+ CCR5+ CEM.NKr cells treated with

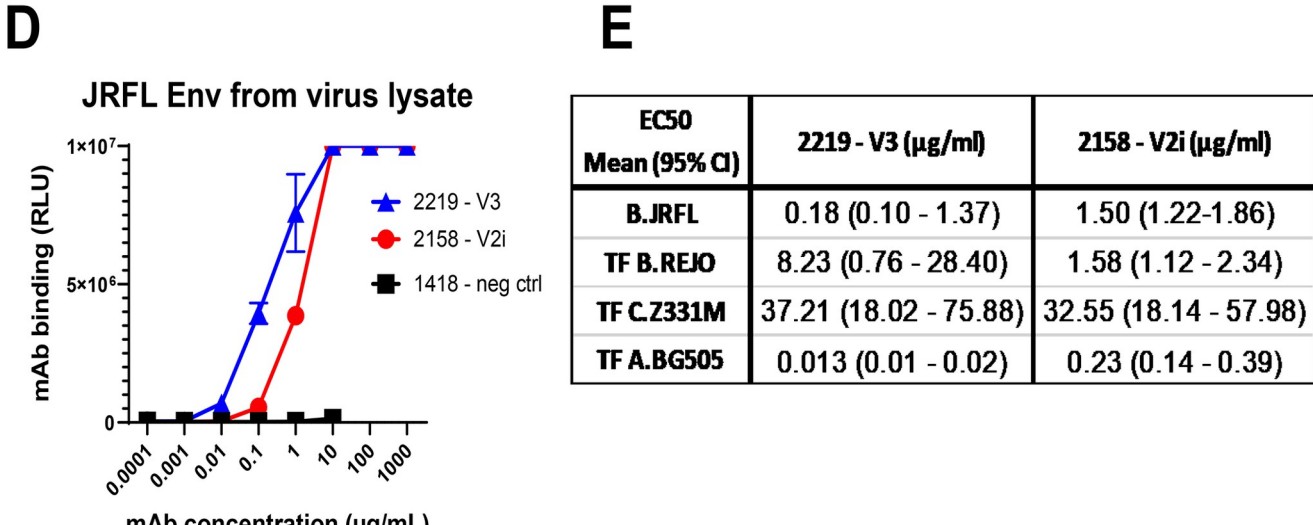

**Fig 2. The binding strength of V2i vs V3 mAb to recombinant JRFL gp120 or virus-derived gp120.** A) Direct ELISA measurement of mAb reactivity to recombinant JRFL gp120 (1 μg/ml coated on plates). B) EC50 values of mAb ELISA binding to recombinant gp120 proteins from different HIV-1 strains. C) Kinetics analysis of mAbs binding to recombinant JRFL gp120 by biolayer interferometry. mAbs were captured by anti-hIgG Fc biosensors and reacted with monomeric gp120 in solution for measurement of Fab-gp120 affinity. Fitted curves (1:1 binding model) are shown in red. Molar concentrations of gp120 are

indicated. D) Measurement of mAb binding to gp120 from virus-derived solubilized JRFL Env captured with ConA in sandwich ELISA. E) EC50 values (mean and range) of mAb binding to Env derived from JRFL and other HIV-1 strains. HIV-1 clade or CRF is indicated by letters before the strain name. TF: transmitted founder.

recombinant JRFL gp120, similar to other V3 mAbs 2557 and 391/95 (**Fig 3D**). In contrast, 2158 recognized neither JRFL Env expressed on cell surface of transfected 293T cells nor JRFL gp120 tethered on the surface of CD4+ cells (**Fig 3C and 3D**). Higher levels of 2219 vs 2158 mAb binding were also detected using a Jurkat cell system previously used to characterize the binding profiles of anti-HIV-1 IgG monoclonal and polyclonal Abs that bind native-like HIV-1 Env [37]. Using this system, we compared the levels of mAb binding to JRFL, NL4.3, and three clade B transmitted founder virus Envs (QH0692; WITO4160; RHPA4259) (**Fig 3E**). For these experiments, 2219 and 2158 were compared with a parvovirus-specific control mAb (1418), another V3-binding mAb (447-52D), and the CD4 binding site targeting mAb b12. We observed that 2219 bound to all of HIV-1 envelopes tested, except NL43, and displayed a similar binding pattern to the other V3 mAb tested, 447-52D (**Fig 3E**). In contrast, 2158 recognized JRFL at detectable but lower levels than 2219. For most of the HIV-1 Envs tested, the level of 2158 binding was comparable to that of the irrelevant anti-parvovirus negative control mAb 1418 (**Fig 3E**).

## V3 mAb 2219, but not V2i mAb 2158, exhibits weak and delayed virus-neutralizing activity

The V3 crown and V2i mAb epitopes are often occluded in functional Env trimers of most HIV-1 strains. However, due to Env structural flexibility, access to these epitopes can occur following extended incubation time to result in virus neutralization [38,39]. When we tested 2219 and 2158 against JRFL IMC, no neutralization was observed for either mAb following a 1-hour virus-mAb incubation (**Fig 3F**). However, when the incubation time was extended to 24 hours, >50% neutralization was achieved by 2219. The neutralizing potency was >3 log10 weaker than the CD4-binding site (CD4bs)-specific bNAb NIH45-46 tested under the same condition. Virus neutralization was not achieved with 2158 even after a 24-hour incubation. Hence, unlike V2i mAb 2158, V3 mAb 2219 had detectable, albeit delayed, neutralizing activity against JRFL IMC. This result is in accord with the greater Fab-mediated capacity of 2219 as demonstrated by virion capture and Env binding on the cell surface (**Fig 3A–3E**). Notably, a concentration of 30 μg/ml, which is equivalent to the IC50 value of 2219 (**Fig 3F**), was maintained in plasma over the experimental period (**Figs 1 and S1**). Nonetheless, in addition to this weak neutralization capacity, non-neutralizing activities are likely to also play a role in the virus control observed with V3 mAb 2219 in the humanized mouse experiments.

## V3 mAb 2219 mediates higher levels of FcγIIa activation, ADCP, and C1q binding as compared to V2i mAb 2158

To determine the Fc-dependent functions mediated by V3 mAb 2219 and V2i mAb 2158, we first measured the capacity of these mAbs to activate FcγRIIa and induce ADCP. FcγRIIa is the primary Fc-receptor that activates ADCP in monocytes and macrophages in response to IgG-opsonized antigens [40]. We employed an FcγRIIa signaling assay with JRFL Δvpu that has previously been used to quantify differences in FcγR activation in HIV-1 controllers [37]. Using this assay, we detected a higher capacity of 2219 to induce FcγRIIa-mediated signaling as compared to 2158 (**Fig 4A**). To assess the ADCP activity of these mAbs, we used gp120-coated beads and FcγRIIa+ THP-1 cells as phagocytes [31,41]. In line with the FcγRIIa

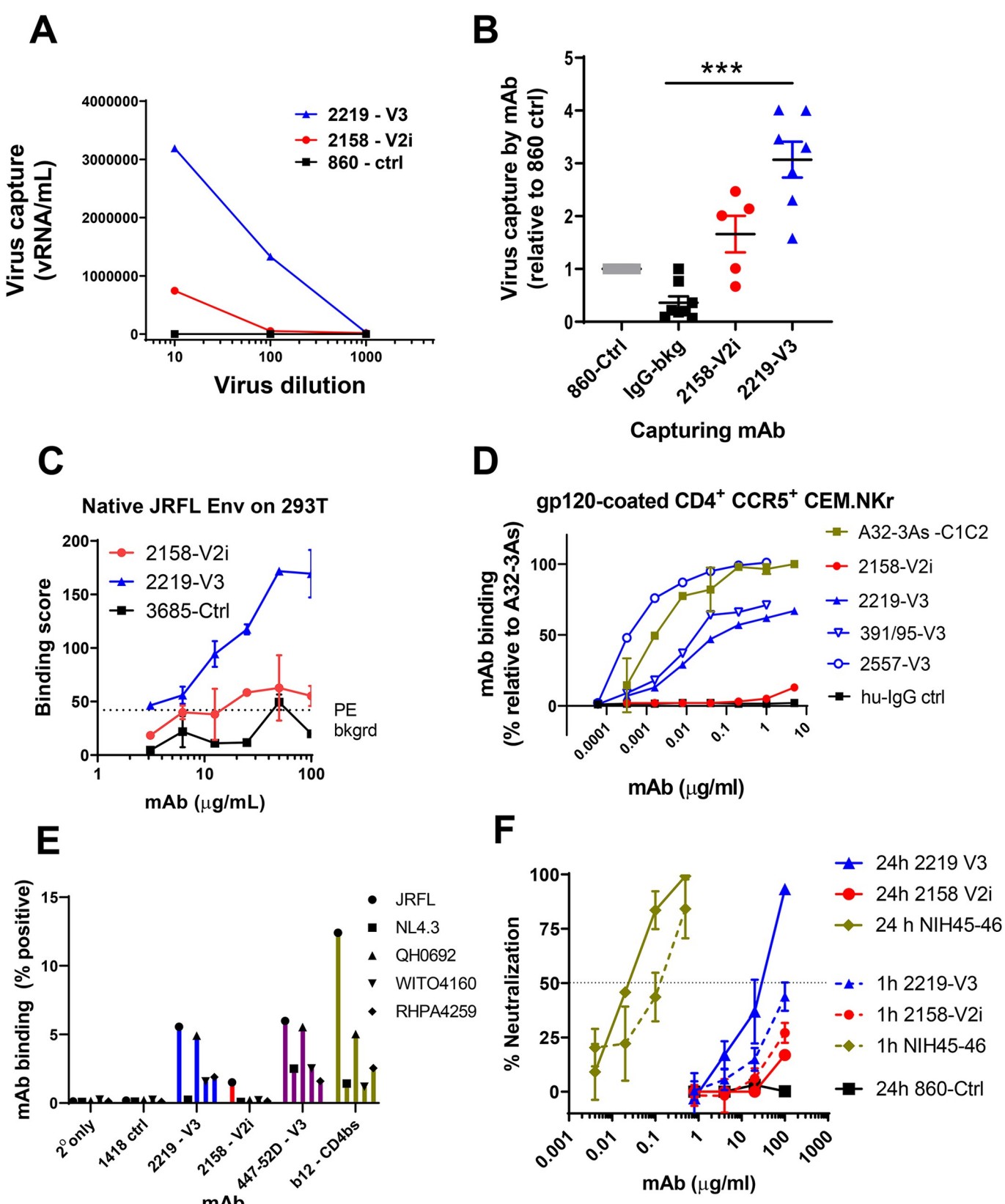

**Fig 3. Capacity of V2i mAb 2158 and V3 mAb to bind and neutralize virus.** A) Capture of JRFL IMC virions by 2158 vs 2219 with titrated amounts of virus input. Irrelevant mAb 860 was used as negative control. Data from one representative experiment. B) Capture of JRFL IMC virions by 2158 vs 2219 at a fixed virus input (5 to 6.5 log10 vRNA copies). Purified human HIV-seronegative IgG and mAb 860 were used as negative controls. Virus capture was calculated relative to that of control mAb 860 (set to 1). Data from 2–3 experiments are shown. ***, p <0.001 by Kruskal-Wallis test with Dunn's multiple comparison. C) Binding of 2158 vs 2219 to native JRFL Env on transfected 293T cells. Irrelevant control mAb 3685 was used as negative control. D) Binding of 2158 vs 2219 to recombinant gp120-coated CD4+ CEM.NKr cells. V3 mAbs 391/95 and 2557 were tested for comparison. MAb A32-3As and human IgG were used as positive and negative controls, respectively. The relative level of mAb binding was calculated based on the binding of A32-3As at 10 μg/ml (set at 100%). E) MAb binding to JRFL and other HIV-1 strains (all clade B) produced and tethered on Jurkat cells. MAbs 2158 and 2219 were compared with V3 mAb 447 and CD4bs mAb b12. HIV-1 Δvpu constructs bearing an mCherry reporter gene were used to transfect tetherin-hi+ Jurkat cells. F) Neutralization of JRFL IMC by 2158 vs 2219 after virus-mAb preincubation for 1 hour or 24 hours using TZM.bl target cells.

signaling assay, we detected higher levels of ADCP activity mediated by 2219, as compared to 2158 (**Fig 4B**). However, both 2219 and 2158 had ADCP activity above the negative control mAb.

We also evaluated complement binding to immune complexes made with gp120 and 2219 or 2158. For this, C1q deposition was measured as the first step in the classical mAb-dependent complement cascade. Both immune complexes made with 2219 and with 2158 showed dose dependent C1q deposition (**Fig 4C**). A slightly higher C1q binding activity was observed with 2219 vs 2158 in agreement with the gp120-binding $EC_{50}$ values of these mAbs (**Fig 2A and 2B**).

## V3 mAb 2219 and V2i mAb 2158 lack the ability to activate FcγRIIIa signaling and induce ADCC

We subsequently examined ADCC, an Fc-dependent activity that also has been shown to play a role in controlling viral infection where NK cells primarily mediate the killing of virally infected cells [42]. ADCC is initiated when an effector cell expressing FcγRIIIa engages with an infected target cell that has been opsonized with virus-specific IgG. To examine the ability of 2219 and 2158 to activate FcγRIIIa, we used an FcγRIIIa signaling assay with JRFL Δvpu and observed that both 2219 and 2158 failed to induce any significant FcγRIIIa signaling in response to virus-infected target cells (**S3A Fig**). In contrast, the CD4bs mAb, b12, efficiently induced FcγRIIIa activation in an Ab-dependent manner in response to virus-infected target cells.

To investigate these findings further, we examined the capacity of 2219 and 2158 to induce ADCC using three different assay systems. These assays provide the opportunity to examine ADCC activity in the context of three distinct pairs of Env-bearing target cells and effector cells. As expected for V2i mAb 2158 that had no binding activity to cell-associated gp120 or full length Env (**Fig 3C, 3D and 3E**), no ADCC activity was detected in each of the three different assays (**S3B**, **S3C and S3D Fig**). However, V3 mAb 2219 also displayed no detectable ADCC against JRFL gp120-coated CEM-NKr-CCR5 cells (**S3B Fig**), even though binding to these target cells was readily detected (**Fig 3D**). The same result was observed with two other V3 mAbs: cradle-type 2557, similar to 2219, and ladle-type 391/95. MAbs 2219 and 2158 also failed to mediate ADCC against SHIV-SF162P3-infected NKR24 reporter cells (**S3C Fig**). In the third assay, which utilizes full-length JRFL IMC-infected primary CD4 T cells, 2219 again showed no binding and no ADCC (**S3D Fig**).

Since the HIV-1 accessory proteins Vpu and Nef are known to impede ADCC responses by controlling Env accumulation at the surface of infected cells and limiting the Env-CD4 interaction that exposes the V3 and other CD4-induced (CD4i) epitopes [43], we examined JRFL IMC lacking Nef and Vpu for comparison. Similar to all other V3 mAbs tested (19b, GE2-JG8, 2424, 2557, 3074, 447-52D, and 268D), 2219 recognized JRFL IMC-infected cells and exhibited ADCC activity when Nef and Vpu were deleted (**S3D Fig**). In contrast, the binding and ADCC

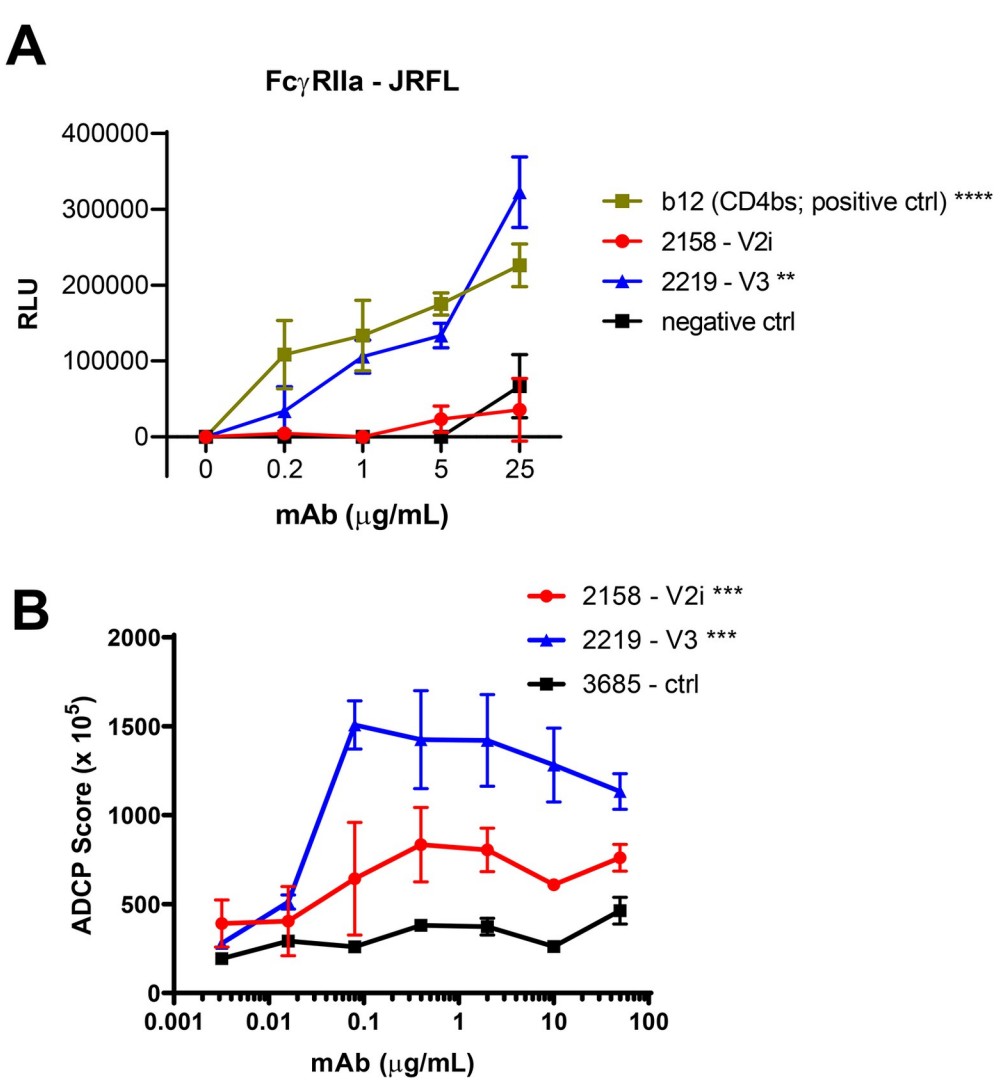

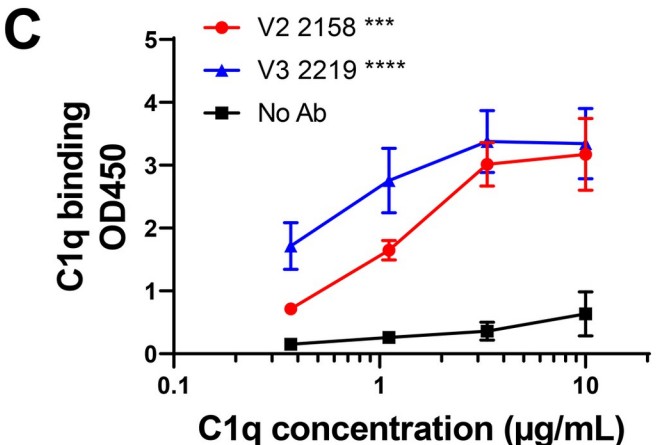

**Fig 4. FcγRIIa signaling, ADCP, and complement binding activities of V2i mAb 2158 vs V3 mAb 2219.** A) FcγRIIA signaling was measured by co-incubating JRFL Δvpu-nucleofected Jurkat cells with Jur-γRIIa luciferase reporter cells in the presence of V2i, V3, or control mAbs. CD4-binding site mAb b12 served as a positive control.

RLU: relative light unit. ****, p <0.0001; **, p <0.01 by two-way ANOVA vs 2158 and negative control. B) ADCP activity was measured using gp120 ZM109-coated beads and THP-1 phagocytic cells in the presence of titrated amounts of mAbs. Mean and SEM from 2 repeat experiments are shown. ***, p <0.001 by two-way ANOVA for 2158 vs 2219 and for 2158 and 2219 vs negative control. C) C1q binding was measured using an ELISA-based assay in which mAbs were reacted with gp120 on ELISA plates and then treated with titrated concentrations of C1q. C1q binding was detected using anti-C1q antibodies and an alkaline phosphate-conjugated secondary antibody. Mean and SD of replicate wells from a representative experiment are shown. **** p <0.0001; *** p <0.001 by two-way ANOVA vs no Ab control.

activity of 2158 was not substantially improved by Nef and Vpu deletions, indicating differential mechanisms regulating the exposure of V2i vs V3 epitopes as previously reported [39,44].

## Significant contribution of V3 mAb 2219 Fc functions against rectal HIV-1 challenge in CD34+ HSC-engrafted humanized mice

Since V3 mAb 2219 showed the capacity to mediate ADCP and complement binding, we sought to examine the contribution of each of these Fc functions in controlling HIV-1 infection in vivo by introducing Fc mutations in the 2219 mAb. A double LALA mutation (L234A/L235A) was made to diminish ADCP and complement activation [45]. Indeed, these Fc changes resulted in significant reduction of ADCP in an assay using the THP-1 phagocytic cells (Fig 5A) and complement activation as measured by C1q and C3d deposition albeit the effect on C1q deposition was partial (Fig 5B). The single KA mutation (K322A), on the other hand, was generated to abrogate C1q and C3d binding without affecting ADCP (Fig 5A and 5B) (45). As expected, we also observed a reduction in ADCP activity with 2219 LALA variant, as compared to the 2219 KA variant and 2219 WT, using an ADCP assay with mouse resident peritoneal mononuclear cells (Fig 5C). The LALA and KA mutations had no effect on the Fab-dependent capacity of 2219, as measured by ELISA reactivity with JRFL gp120 (S4A Fig) and neutralization of JRFL IMC following a 24-hour virus-mAb incubation S4B Fig).

To examine the in vivo effects of these Fc mutations, the 2219 LALA and KA variants were administered to CD34+ HSC-engrafted mice. The wild type 2219 mAb and an irrelevant control mAb 860 were tested in parallel for comparison. The 4 groups of mice (n = 11-16/group from different cohorts) were given mAb (2 x 700 μg/animal, intraperitoneal) and then challenged with JRFL IMC (2x 700 TCID50, intrarectal) (Fig 6 and S1 Table). Plasma virus loads were measured over time and presented as relative AUC over that of the control 860 group in each experiment. While the plasma vRNA loads varied greatly among individual animals, the 2219 WT group had significantly lower plasma virus burden than the control group (Fig 6A). The vRNA AUC values of the LALA and KA groups were not significantly different from the 2219 WT or control group, but there was a trend toward higher vRNA loads especially in the KA mutant group vs the 2219 WT group. Increased viral loads in the KA mutant group were observed even though comparable 2219 WT and KA concentrations were detected in plasma (S4C Fig). This is in agreement with past findings showing equivalent half-lives of b12 WT, LALA, and KA variants upon passive infusion to non-human primates [46].

When the virus loads in splenocytes were measured, we observed that reduced levels of cell-associated vRNA were maintained for the WT and LALA groups vs the control 860 group, but not in the KA group (Fig 6B). However, vDNA levels were unchanged (Fig 6C). Measurement of p24+ cells further revealed lower percentages of p24+ cells among human CD4 T cells in the spleen of mice that received 2219 WT vs control 860 (Fig 6D). Indeed, the 2219 WT group consistently had significantly lower levels of vRNA in plasma and cell-associated vRNA and p24+ CD4 T cells in the spleen than the control 860 group (Fig 6A, 6B and 6D). The percentages of p24+ cells in the LALA and KA groups were higher than that of the 2219 WT group,

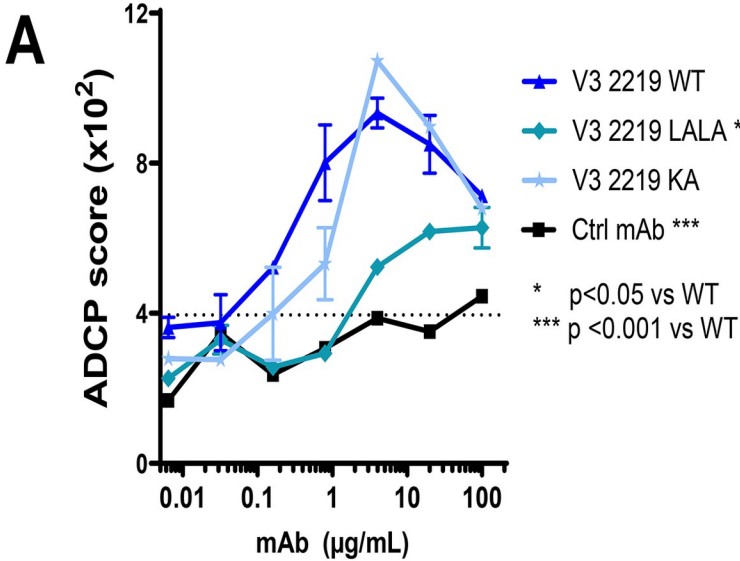

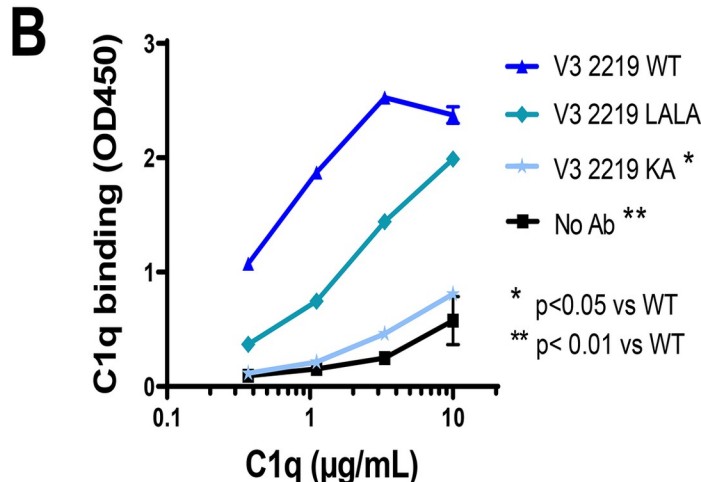

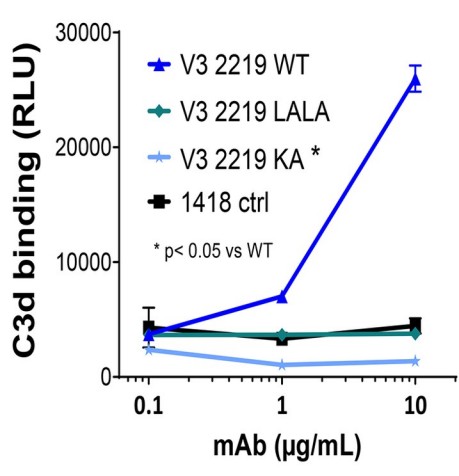

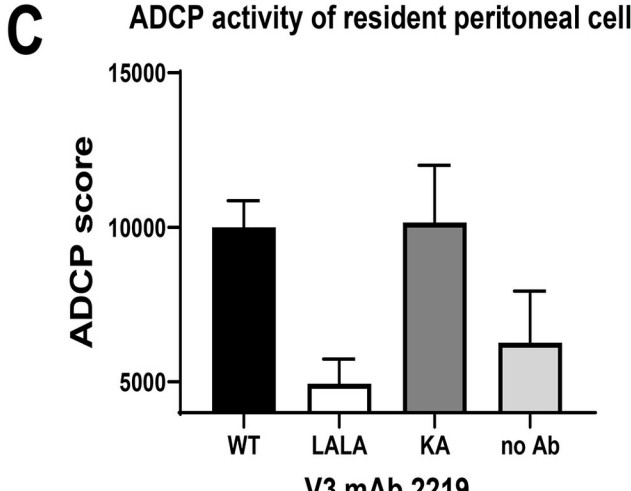

**Fig 5. ADCP and complement binding activity of V3 mAb 2219 with WT or mutated Fc fragments (LALA and KA).** A) ADCP activity was measured using gp120 ZM109-coated beads and THP-1 phagocytic cells in the presence of 2219 WT or Fc mutants. Statistical test was done using two-way ANOVA. B) Complement binding by V3 mAb 2219 with WT and mutated Fc fragments was detected in ELISA or multiplex bead experiments in which C1q or C3d deposition to mAbs in complex with V3 MN peptide was measured with anti-C1q or anti-C3d secondary antibodies. Comparison was analyzed with two-way ANOVA. C) ADCP of 2219 WT and Fc mutants was detected by measuring phagocytosis of mAb-treated gp120 ZM109-coated beads by resident peritoneal macrophages from NSG mice.

but with only 5 animals tested per group in this assay, a significant difference was achieved only between KA vs WT groups. These p24 data recapitulated the pattern seen with plasma vRNA (Fig 6A) and cell-associated vRNA (Fig 6B), although the KA mutation only caused partial reversal and the differences between KA and WT groups were not significant in the latter two measurements, indicating the KA mutation did not completely abrogate the viral control activity of 2219. These data recurrently demonstrate that LALA and KA mutations diminished virus suppressive activity of 2219, but KA had a greater impact than LALA. The differential effects of KA vs LALA corresponded with a greater reduction of complement binding by KA than LALA mutations and contrasted with the minimal effect of KA on ADCP. Together with data in Fig 5, the study indicates the important contribution of Fc-dependent functions, in particular complement activation, to the antiviral potency of V3 mAb 2219.

To examine the ability of V3 mAb 2219 to control virus in cells other than CD4 T cells, we examined p24+ cells among human monocytes (CD3-CD11c-CD14+). p24+ monocytes were readily detected in all groups (Fig 6E). Unlike that seen with p24+ CD4 T cells, no decrease was measurable in the percentages of p24+ monocytes from the 2219 WT vs control groups. Similarly, no change was seen in the LALA and KA groups. These data point to the disparities in the ability of V3 mAb 2219 to control virus infection in different cell types and the lack of antiviral potency against virus reservoirs beyond CD4 T cells.

## Discussion

This paper provides the first evidence for the ability of passively infused V2i mAb 2158 and V3 mAb 2219 to reduce levels of cell-associated vRNA and vDNA in a CD34+ HSC-engrafted humanized mouse model upon challenge with a resistant tier 2 HIV-1, JRFL IMC. Neither mAbs showed detectable neutralizing activity against the challenge virus in the standard TZM.bl neutralization assay, but V3 mAb 2219 displayed a greater capacity than V2i mAb 2158 to bind free virions, cell-associated virions, cell-bound gp120, and membrane-associated Env on transfected or infected cells. Unlike V2i mAb 2158, V2 mAb 2219 was able to exert delayed neutralization against JRFL IMC detectable after prolonged mAb-virus pre-incubation, although the IC50 value was >3 log less potent than bNAbs such as NIH45-46. Correspondingly, V3 mAb 2219 exerted a greater control of virus infection in humanized mice, as indicated with consistent reduction of vRNA in plasma and cell-associated vRNA in tissues. V2i mAb 2158, on the other hand, had minimal effects on plasma viremia and did not reduce virus burden in tissues to the same extent observed with V3 mAb 2219. Notably, neither mAb impacted the vDNA loads. These data suggest that 2219, with its weak and delayed neutralizing activity, cannot prevent or reduce infection but may contribute to protection by suppression of virus replication. The results also indicate the potential involvement of non-neutralizing functions in the anti-HIV-1 suppressive mechanisms wielded by the mAb.

As compared to V2i mAb 2158, V3 mAb 2219 showed a greater capacity to activate FcɣRIIa signaling upon binding to cell surface Env and induce FcɣRIIa+ THP-1 phagocytic cells to mediate ADCP of Env-coated beads. Upon immune complex formation, 2219 also was able to engage C1q, the initial step in the classical complement cascade. The FcɣRIIa- and complement-mediated functions were reduced by the introduction of Fc mutations, and these

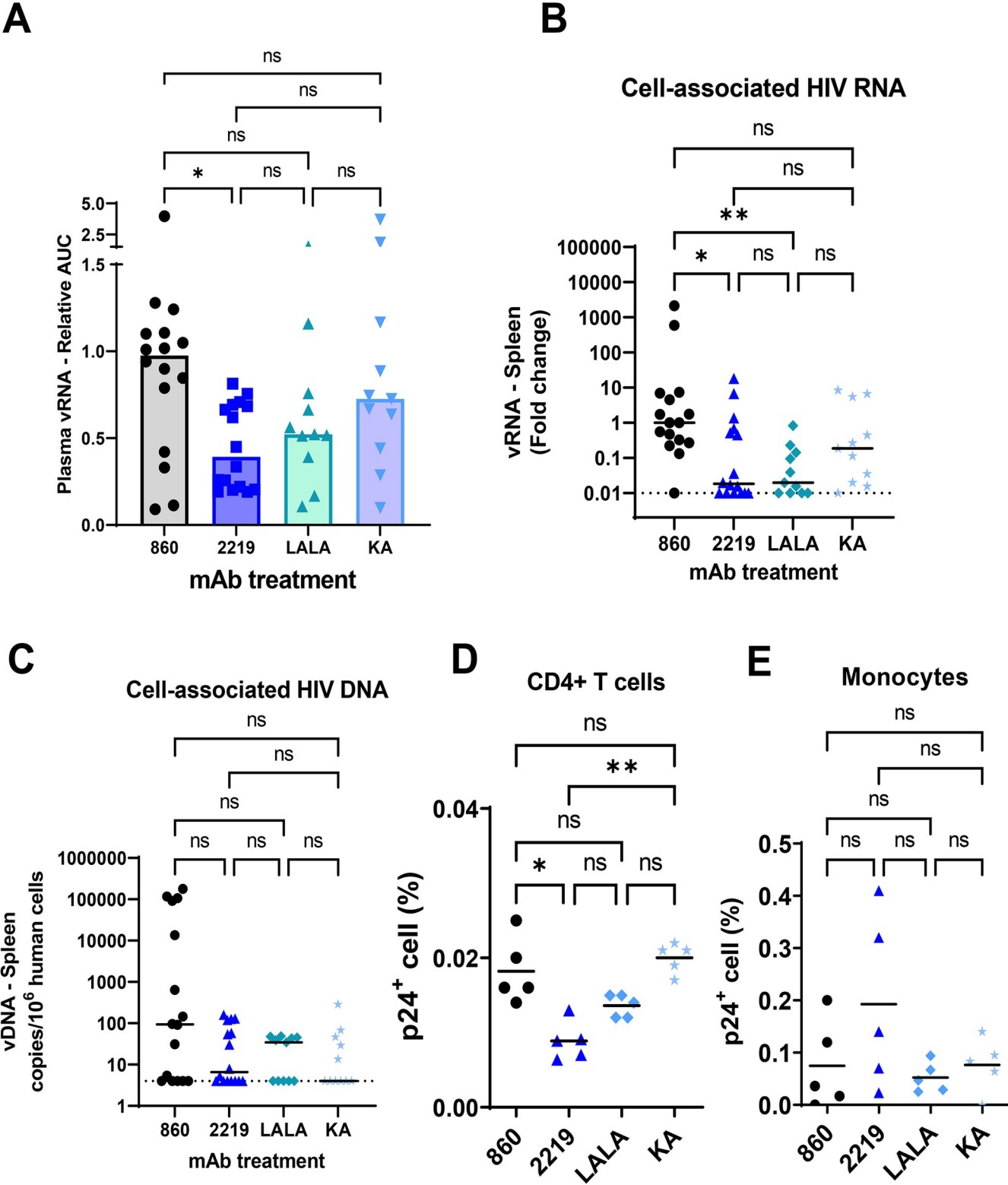

**Fig 6. Virus control by passively administered 2219 WT vs LALA vs KA in humanized mice challenged with JRFL IMC.** A. Plasma vRNA loads as measured by vRNA AUC observed in individual mice that received 2219 WT or LALA or KA variants and challenged with JRFL IMC. Data are presented as relative AUC over mean AUC of the control 860 group included in each experiment. Data are compiled from experiments using 2 or 3 cohorts of mice that were generated with different HSC donors. N = 16, 16, 11 and 11 for the groups treated with 860 (irrelevant control), 2219 WT, LALA, and KA, respectively.

B. Relative levels of cell-associated vRNA in the spleen collected from individual animals on the last day of experiments. Data from 2 or 3 experiments are presented as fold changes over median of the 860 group tested in each experiment. C. Relative levels of cell-associated vDNA in the spleen on the last day of experiments. Data from 2 or 3 experiments are presented as vDNA copies/$10^6$ human CD45+ cells. D-E. Percentages of p24+ cells among CD4 T cells (D) and monocytes (E). Spleen cells collected from mice in one experiment (n = 5/group) were subjected to intracellular staining with anti-p24 mAb KC57 and staining for viability markers, human CD45 (huCD45), mouse CD45 (mCD45), CD4 T cells (CD3+CD8-), and monocytes (CD3-CD11c-CD14+). Analysis was done on viable hCD45+ and mCD45- cells. Cells from mock-infected mice are used as negative controls (S5 Fig). Statistical analysis was performed using Kruskal-Wallis one-way ANOVA test with Dunn's multiple comparison. *, p <0.05; **, <0.01; ns: not significant (p>0.05). Dotted lines: limit of detection. Bars and horizontal lines: median.

mutations partially reversed virus suppression observed in the wild type 2219-treated humanized mice. Of note, the KA mutation that only abrogated complement binding had the same or greater effects as the LALA mutations which decreased ADCP more than complement binding. The greater effects of KA vs LALA observed in vivo corresponded best with a greater reduction of complement binding by KA vs LALA variants, indicating the importance of complement-mediated functions in HIV-1 control by V3 mAb 2219, although virus suppression was far from complete. The partial reversal of virus control seen in the KA and LALA groups indicates that these Fc mutations did not completely abrogate the antiviral activity of 2219; this is likely attributed to virus neutralization, which albeit weak against the JRFL challenge virus was retained in the 2219 Fc variants. Further, the contribution of complement-dependent antibody functions to HIV-1 control remains unclear. The two complement assays in this study examined C1q and C3d deposition on mAbs bound to gp120 on solid surfaces; the impact of complement on antibody-coated virions or cells in vitro and in vivo needs further investigation. It is also unknown if the importance of complement functions seen with V3 mAb 2219 can be generalized to other non-neutralizing mAb or detected with other challenge viruses and in a different animal model, as disparate results have been reported from the in vivo prophylactic testing of bNAbs. For example, higher rates of virus infection were observed upon administration of the Fc variants of bNAbs (e.g. anti-CD4bs 3BNC117 and anti-V3 glycan PGT121) with diminished versus intact Fc functions to reporter mice transduced to express human CD4 and CCR5 receptors and challenged with HIV-1 [47]. In the non-human primate models, passive transfer of PGT121 and its LALA mutant equally protected against intravenous or rectal SHIV challenge [48–50], whereas anti-CD4bs bNAb b12 required FcγR binding activity for optimal protection [46]. Hence, it is likely that the contribution of FcγR engagement and its downstream antiviral functions varies among mAbs, depending on epitope and neutralization potency. FcγR effector functions may play a greater role in reducing HIV-1 burden in the context of weaker neutralization potency.

In contrast to their capacity to trigger FcγRIIa signaling and mediate ADCP, neither 2158 nor 2219 had measurable FcγRIIIA signaling and ADCC activity. Our study utilized four different assays performed with distinct target cells, effector cells and readouts and yielded consistently negative results. Although diverging results may be observed in other assays under yet different conditions, our results are in accord with past data showing that the ADCC activity detected in a modified rapid fluorescent ADCC assay with gp140-treated CEM-NKr cells and NK effector cells was <10% for both mAbs [51]. In another study that utilized virus-infected primary CD4 T cells as target cells and NK effector cells, V2i mAbs also showed no or weak ADCC against tier 2 viruses, while the ADCC activities against tier 1 viruses were more robust [52]. In an ADCC assay that utilized CD4 T cells infected with JRFL IMC as target cells vs the Nef- and Vpu-deleted counterpart, the failure of V3 mAb 2219 to recognize and elicit ADCC against JRFL IMC-infected target cells was overturned by Nef and Vpu deletion. Among the manifold effects of Nef, the downregulation of CD4 has been attributed to minimizing CD4-induced epitope exposure, while the reduction of NKG2D ligand expression increases

the resistance of HIV-1-infected cells to ADCC [53–56]. Apart from contributing to CD4 downregulation, Vpu promotes virus release from infected cells by counteracting the restriction factor tetherin (BST2), thus limiting accumulation of virions and Env antigens on the cell surface and in turn reducing their detection by antibodies [37,54,57]. In contrast to the conspicuous effects of Nef and Vpu deletion on V3, the exposure of V2i epitopes was not affected, consistent with our earlier findings implicating distinct mechanisms in masking V3 vs V2i epitopes [39,44]. Nonetheless, 2219 also failed to elicit ADCC against gp120-treated CD4+ CEM. NKr cells where V3 exposure was augmented by gp120-CD4 interaction and a high binding level of 2219 to these cells was observed. In this case, the lack of ADCC activity can be attributed to 2219's inefficient induction of FcɣRIIIa signaling, a requisite for effector cells to kill target cells via the ADCC mechanism. The reason for this observation remains unknown. MAbs specific for the V3 crown interact with their epitopes via the cradle or ladle binding modes, allowing different angles of approach [23,24]. However, the binding modes did not appear to affect ADCC capability. When we tested the cradle-type V3 mAbs 2219 and 2558 and the ladle-type V3 mAb 391/95, all of which were expressed as recombinant IgG1, we detected their binding to the gp120-coated CD4+ CEM.NKr target cells but all three mAbs lacked ADCC activities.

Measurement of p24+ cells among different cells susceptible to HIV-1 infection in the humanized mouse model indicates a limitation of V3 mAb 2219 in controlling infection beyond CD4 T cells. To the best of our knowledge, our study is the first to reveal the differential potency of V3 mAb 2219 against HIV-1 infection in CD4 T cells vs monocytes. Mimicking HIV-1-infected humans, virus-infected CD34+ HSC-engrafted humanized mice harbor HIV-1 in various cell types in the blood and the lymphoid tissues [58–60]. However, few studies evaluating neutralizing and non-neutralizing antibody functions have considered the influence of target cell types on the antibody potency. In a 2014 report, Lederle et al. [61] demonstrated that potent bNAbs against different Env epitopes, including VRC01, PGT121, 10–1074, 2G12, b12, 4E10, and 2F5, displayed distinct inhibitory activity when plasmacytoid dendritic cells (pDC) vs monocyte-derived DCs (moDC) were used as target cells. Interestingly, higher concentrations were consistently required for all tested mAbs to inhibit 90% infection of pDC vs moDCs target cells. Moreover, even though a relatively sensitive tier 1 HIV-1 BaL was tested, these IC90 values were much higher than those observed in the standard neutralization assay with TZM.bl target cells [62,63]. A similar pattern was evident for non-neutralizing mAbs such as anti-gp41 mAbs 246-D and 4B3, which showed weak but detectable inhibitory activity in moDCs but not in pDCs [61]. The mechanistic explanations for this phenomenon are yet undefined. Whether such differential resistance impacted the capacity of 2219 to control HIV-1 in CD4 T cells vs other cell types is unknown and needs further investigation. Future studies to evaluate the Env expression and Env-antibody interaction on various primary cells infected with HIV-1 are warranted.

The humanized mouse experiments in this study were designed to evaluate the prophylactic effects of V2i mAb 2158 and V3 mAb 2219 against mucosal HIV-1 exposure. Considering the absence of potent neutralizing activity, it was not surprising that these mAbs did not confer sterilizing immunity. We noted that the dynamics and levels of virus loads in this mouse model varied greatly, reflecting variability among individual animals and different HSC donors. The levels of human CD45+ cell reconstitution also varied but did not predict blood and tissue virus loads in animals with or without mAb treatment. Still, a consistent pattern of virus load reduction was detectable in the V3 mAb 2219-treated mice. It is worth noting, however, that the experimental system had several limitations, the most prominent being the high dose of challenge virus required to establish infection via a rectal route in this model. Two inoculations of 700 TCID50 were given to each animal based on a prior titration showing that

a single inoculation yielded infection only in a fraction of the animals. This and other experimental models, e.g. the titrated multiple challenges used to infect rhesus macaques with 3 to 5 intrarectal exposures, do not reflect the transmission efficiency of HIV-1 in humans, which is estimated at a frequency of 1 to 8 transmissions per 1000 exposures [64–66]. The challenge virus was a chimeric JRFL-NL4.3 infectious clone lacking Vpr, a viral protein important for virus spread and pathogenesis in vivo and for virus replication in myeloid cell populations [67]. It should also be noted that we generated humanized mice using the NSG strain that lacks the C5 complement component needed to create the membrane attack complex for virion or cell lysis [68]. Yet, independent of C5, the upstream classical complement cascade that starts from C1q binding to antibody-antigen immune complexes remains operative, producing anaphylatoxin and opsonins (C3a, C3b, iC3b, and C3d) capable of engaging G protein-coupled (C3AR1) and complement (CR1 and CR2) receptors. Indeed, the lack of virus control by the 2219 KA mutant, which is unable to bind C1q and generate C3d, implies a role of the C3 complement activity in controlling HIV-1 infection. Another caveat is that in humanized mice, there are mouse effector cells that participate in ADCP and complement-dependent functions but are not susceptible to HIV-1 infection. By contrast, in the context of non-sterilizing protection, human effector cells can be infected and, as a result, may have compromised effector functions that diminish the antibody effectiveness against the virus.

Altogether, this passive transfer study using a humanized mouse model demonstrates the ability of V2i and V3 mAbs to exert control of HIV-1 in the absence of potent neutralization and ADCC activity. Instead, ADCP and complement-mediated functions play a role in suppressing virus infection in CD4 T cells, although virus control was not achieved in other cell types such as monocytes which also harbor the virus. As V2i and V3 crown are representatives of highly immunogenic epitopes that are targeted by antibodies readily elicited by vaccination, data from this study offer further evidence that non-neutralizing activities mediated by antibodies will be important to induce with vaccines designed to prevent and control HIV-1 infection.

## Materials and methods

### Ethics statement

Animal work was reviewed and approved by the University of North Carolina at Chapel Hill IACUC (ID: 17–051.0-B).

### Monoclonal antibodies

V2i mAb 2158 and V3 mAb 2219 were produced as recombinant IgG1 in transfected 293F cells, affinity purified by protein A (HiTrap, Sigma-Aldrich), and tested for endotoxin levels (GenScript) prior to use in experiments. The complete plasmid sequencing showed that the Fc heavy chain domains of these mAbs were identical. The KA and LALA mutations were introduced to the Fc fragment of 2219 by QuikChange II XL Site-Directed Mutagenesis Kit (Agilent Technologies) according to the instruction manual, and confirmed by sequencing. Like the wild type counterpart, the KA and LALA variants were produced in 293F cells following transfection.

For comparison, other V2i and V3 mAbs were included in some in vitro assays, whereas anti-parvovirus mAb 860-50D (designated as 860 herein) and CD4bs-specific bNAb NIH45-46 were used as negative and positive control, respectively. These mAbs were also produced in 293F cells.

## Humanized mouse experiments

Humanized mice were generated as described [59] by injecting human CD34+ hematopoietic stem cells (HSCs) from fetal liver tissues into the liver of irradiated NOD-SCID IL2RɣNULL (NSG) neonates (80 rads, 1 to 5 days old, $0.2 \times 10^6$ cells/animal). Human fetal livers were obtained from medically indicated or elective termination of pregnancies through a non-profit intermediary working with outpatient clinics (Advanced Bioscience Resources, Alameda CA). Human leukocyte engraftment was monitored at week 12 after transplantation by hu-CD45 + staining. Each experiment used male and female animals from the same cohort. The levels of hu-CD45+ reconstitution at week 12 after transplantation are shown in **S1 Table**. Each animal received intraperitoneal injection of mAb (2x 700 µg/animal) and was challenged intrarectally with 2x 700 TCID50 (equivalent to a total of 5000 pfu) of JRFL IMC, an infectious molecular clone (NFN-SX-r-HSAS) with chimeric JRFL-NL4.3 Vpu and Env and vpr-deleted NL4.3 backbone (a gift from Dr. J. A. Zack, UCLA) [35]. The intraperitoneal route allowed delivery of a bolus amount of mAb in a 0.5 to 1.0 ml volume, while the intrarectal virus administration was designed to test a mucosal exposure applicable for both male and female. Blood and tissue collection times for each experiment are shown in **S1 Table**. Plasma vRNA was extracted by QIAamp Viral RNA Mini Kit (QIAGEN) and quantified by real time PCR (ABI Applied Biosystem) as in [58]. Tissues were collected at the end of experiment, nucleic acid was extracted and cell-associated vDNA and vRNA levels were measured as described previously [58,59,69]. Virus-infected cells were detected by flow cytometry following mAb staining against intracellular p24 and cellular markers (CD3, CD8, CD4, CD25, FoxP3, CD14, CD11c, HLA-DR, CD123).

## ELISA with soluble Env proteins

A direct ELISA was performed as described in [70] to assess mAb reactivity with recombinant gp120 proteins coated on the plates. To examine mAb reactivity with virus-derived Env proteins, a sandwich ELISA was used in which 1% Trixon X-100-treated virus lysates were added to plates pre-coated with ConA (50 µg/ml, Sigma) and reacted with anti-Env mAbs [71]. MAb binding was detected with alkaline phosphate-conjugated antibodies against human IgG or biotinylated anti-human IgG antibodies and horseradish peroxidase-streptavidin.

## Biolayer interferometry

The kinetics analysis of gp120-mAb binding was performed by biolayer interferometry using an Octet Red96 instrument (ForteBio) [70]. mAbs were immobilized on Anti-hIgG Fc Capture (AHC) biosensors and dipped into recombinant JRFL gp120 monomers at the designated concentrations. This experimental condition measured the affinity of each Fab fragment for gp120 in a 1:1 stoichiometry. All samples were diluted in PBS (pH 7.4) supplemented with BSA (0.1% w/v) and Tween 20 (0.02% v/v). A baseline reference, consisting of a loaded AHC sensor run with a buffer blank for both association and dissociation steps, was utilized to correct for drift. Duplicate experiments were performed. After subtracting reference curves, data were analyzed with the Octet Data Analysis software by employing a 1:1 binding model for a global fit analysis of association and dissociation curves.

## Virus capture

MAb binding to virion was assessed by pre-incubating mAb with cell-free virus particles for 24 hours at 37˚C and treating the mAb-virus mixture with protein G-coated magnetic beads (Protein G Mag Sepharose Xtra, Cytivia). The beads were pelleted, washed, and subjected to vRNA

quantification by real time PCR using the Abbott m2000 System according to the manufacturer's instruction.

## MAb binding to Env on cells

Flow cytometry was performed to detect mAb binding to Env on transfected 293T cells [72] or on target cells used in the ADCC assays [31,54,73,74]. Fluorescent secondary antibodies against human IgG were used for detection of mAb binding.

## Virus neutralization

Neutralizing activity of anti-Env mAb was measured as a reduction in β-galactosidase reporter gene expression of TZM-bl target cells. Neutralization was performed with the standard 1 hour mAb-virus incubation or the prolonged 24 hour incubation as described [72].

## FcγR signaling

FcγR signaling was measured according to a published protocol [37] using Jurkat cell–derived reporter cell lines (Jur-γRIIa and Jur-γRIIIa) that contain an integrated NFAT-driven firefly luciferase reporter gene. The Jur-γRIIa or Jur-γRIIIa cells were co-cultured for 16 hours at a 2:1 ratio with HIV-1 Δvpu-infected tetherin$^{high}$ CD4+ lymphocytes that were pre-treated with each mAb for 15 minutes. The luciferase activity was measured using a luciferase assay kit (Promega) and subtracted with the background level obtained from co-cultures in the absence of mAb.

## ADCP

The ADCP assay was performed as described [31] using gp120-coated fluorescent NeutrAvidin beads (1-μm diameter) and THP-1 cells or resident peritoneal macrophages from NSG mice. Beads pre-treated with mAbs were added to THP-1 cells, and phagocytosis was measured by flow cytometry after an overnight incubation. ADCP scores were calculated as: (percentage of bead-positive cells × mean fluorescence intensity of bead-positive cells).

## Complement deposition

C1q binding to immune complexes made with V2i mAb 2158 or V3 mAb 2219 was measured in ELISA. MAb was reacted with antigen coated on the plates and then treated with serially diluted C1q from human serum (Sigma). C1q binding was detected with horseradish peroxidase-conjugated anti-C1q antibody. C3d deposition was detected using Luminex assay according to a published protocol [75,76] with some modifications. Antigen-coupled xMAP beads were incubated with serially diluted mAb, and then treated with human complement serum (33.3%, Sigma) at 37°C for 1 hour. C3d production and deposition was measured by biotinylated anti-C3d mAb (Quidel) and PE-streptavidin.

## ADCC

The ADCC assay using gp120-coated CD4+ CEM.NKr target cells and PBMCs as effector cells was done as described in [73], whereas the assay with virus-infected NKR24 reporter cells and human NK cell line KHYG-1 was done according to [74]. The third ADCC assay was performed using full length HIV-1 JRFL IMC-infected primary CD4 T cells and PBMC effector cells [54,77].

## Supporting information

**S1 Fig. Pharmacokinetic study of human V2i mAb 2158 and V3 mAb 2219 in NSG mice.**
Each mice was given V2i mAb 2158 and V3 mAb 2219 (700 μg per mAb) intraperitoneally.
Plasma was collected and monitored for the concentration of V2i mAb 2158 and V3 mAb
2219 from day 0 to day 14 using ELISA with V1V2-1FD6 or peptide V3 antigens and the
respective mAbs as standard. Based on the average values, the half life was estimated to be >14
days for V2i mAb 2158 and 11 days for V3 mAb 2219.
(PDF)

**S2 Fig. Reduced levels of plasma vRNA and cell-associated vRNA and vDNA in CD34+
engrafted humanized mice treated with V2i or V3 mAbs and infected with JRFL IMC.** A)
Plasma vRNA loads from day 0 to day 9 in mice that received control mAb 860, V2i mAb
2158, or V3 mAb 2219. Individual mouse data from Fig 1B are shown. B) Relative levels of
cell-associated vRNA in bone marrow and mesenteric lymph nodes collected at day 9 from
animals in panel A. C) Relative vDNA levels in bone marrow and mesenteric lymph nodes
from animals in panel A. *, p <0.05 by Kruskal-Wallis one-way ANOVA test with Dunn's
multiple comparison. Significant differences are marked, the other comparisons show no sig-
nificant difference.
(PDF)

**S3 Fig. V2i mAb 2158 and V3 mAb 2219 lack the ability to mediate FcγRIIIa signaling and
ADCC activity.** A) FcγRIIIA signaling was measured by co-incubating JRFL Δvpu-nucleo-
fected Jurkat cells with Jur-γRIIIa luciferase reporter cells in the presence of V2i, V3, or control
mAbs. CD4-binding site mAb b12 served as a positive control. RLU: relative light unit. B)
ADCC activity was determined using CD4$^+$ CEM.NKr target cells that were coated with
recombinant gp120 JRFL, treated with V2i, V3, or control mAbs, and incubated with PBMCs
as effector cells. V2i mAb, cradle-type V3 mAbs 2219 and 2557, and ladle-type V3 mAb 391/95
were tested along with anti-C1C2 mAb A32-3As (positive control) and purified human IgG
(negative control). C) ADCC activity was examined in a second assay in which NKR24 lucifer-
ase reporter cells were infected with virus for 3 to 4 days, combined with the effector cells,
human CD16$^+$ NK cell line KHYG-1, at an effector-to-target cell ratio of 5:1, and incubated
with serially diluted mAbs for 8 hours. The NKR24 target cell viability was measured by lucif-
erase activity. PGT121 was used as a positive control. D) ADCC activity was also determined
against primary CD4+ T cells infected with JRFL IMC or the Nef and Vpu-deleted counter-
part. Binding of 2158 and 2219 to virus- vs mock-infected target cells was first examined by
flow cytometry (top). Other V2i and V3 mAbs were tested for comparison. 3BNC117 (anti-
CD4bs) and A32 (anti-C1C2) served as positive controls. ADCC were subsequently measured
with PBMC effector cells at an effector-to-target ratio of 10:1 (bottom). Viability dye was used
to measure cytotoxicity against infected target cells. Each mAb was tested at 5 μg/mL. E) Sum-
mary of experimental parameters and conditions for FcγRIIIA signaling and ADCC assays in
Panels A-D.
(PDF)

**S4 Fig. Fc mutations LALA and KA do not alter gp120 binding, neutralizing activity, or
plasma concentration of V3 mAb 2219.** A) ELISA reactivity of 2219 WT vs Fc mutants was
tested against recombinant gp120 JRFL. B) Neutralization of JRFL by 2219 WT vs Fc mutants
was assessed with TZM.bl target cells after 24-hour mAb-virus incubation. CD4bs-specific
bNAb NIH45-46 and irrelevant mAb 860 were included as controls. C) Concentration of 2219
WT vs Fc mutants in plasma of mice after passive infusion with each mAb (700 μg x 2 doses/

animal, intraperitoneal, days 0 and 2).
(PDF)

**S5 Fig. Flow cytometry for detection of p24+ cells in the spleen of humanized mice treated with 2219 WT or Fc mutants and challenged with JRFL IMC.** Spleen cells were subjected to intracellular staining with anti-p24 mAb KC57 and staining for markers of cell viability, human CD45 (huCD45), mouse CD45 (mCD45), CD4 T cells (CD3+CD8-), and monocytes (CD3-CD11c-CD14+). p24+ cells were detected in CD4 T cells or monocytes gated from viable human cells (hCD45+ and mCD45-). Dot plots from representative animals in the treated and mock groups are shown.
(PDF)

**S1 Table. Passive transfer experiments with V2i and V3 mAbs in human CD34+ HSC-engrafted mice.**
(PDF)

## Acknowledgments

The authors thank Dennis Burton for the infectious molecular clone JRFL used in the ADCC assay using primary CD4+ T cells and Hans Watanabe for graphic assistance.

## Author Contributions

**Conceptualization:** Catarina E. Hioe.

**Formal analysis:** Catarina E. Hioe, Guangming Li, Ourania Tsahouridis, Xiuting He, Masaya Funaki, Jéromine Klingler, Alex F. Tang, Roya Feyznezhad, Daniel W. Heindel, Xiao-Hong Wang, David A. Spencer, Guangnan Hu, Namita Satija, Jérémie Prévost, Andrés Finzi, Ann J. Hessell, Shixia Wang, Shan Lu, Benjamin K. Chen, Susan Zolla-Pazner, Chitra Upadhyay.

**Funding acquisition:** Catarina E. Hioe, Andrés Finzi, Ann J. Hessell, Shixia Wang, Shan Lu, Benjamin K. Chen, Susan Zolla-Pazner, Chitra Upadhyay, Lishan Su.

**Investigation:** Guangming Li, Xiaomei Liu, Ourania Tsahouridis, Xiuting He, Masaya Funaki, Jéromine Klingler, Alex F. Tang, Roya Feyznezhad, Daniel W. Heindel, Xiao-Hong Wang, David A. Spencer, Guangnan Hu, Namita Satija, Jérémie Prévost, Raymond Alvarez.

**Methodology:** Guangming Li, Andrés Finzi, Ann J. Hessell, Shixia Wang, Shan Lu, Benjamin K. Chen, Susan Zolla-Pazner, Chitra Upadhyay, Raymond Alvarez.

**Project administration:** Catarina E. Hioe.

**Resources:** Guangming Li, Xiaomei Liu, Andrés Finzi, Ann J. Hessell, Shixia Wang, Shan Lu, Benjamin K. Chen, Susan Zolla-Pazner, Chitra Upadhyay, Raymond Alvarez, Lishan Su.

**Supervision:** Catarina E. Hioe.

**Writing – original draft:** Catarina E. Hioe.

**Writing – review & editing:** Catarina E. Hioe, Guangming Li, Raymond Alvarez, Lishan Su.

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
