## [Decision Letter · Decision Letter 0]

16 Nov 2021

Dear Dr. Hioe,

Thank you very much for submitting your manuscript "Non-neutralizing antibodies targeting the immunogenic regions of HIV-1 envelope reduce mucosal infection and virus burden in humanized mice" for consideration at PLOS Pathogens. As with all papers reviewed by the journal, your manuscript was reviewed by members of the editorial board and by several independent reviewers. The reviewers appreciated the attention to an important topic. Based on the reviews, we are likely to accept this manuscript for publication, providing that you modify the manuscript according to the review recommendations.

Sincerely,

Daniel C. Douek

Associate Editor

PLOS Pathogens

Richard Koup

Section Editor

PLOS Pathogens

Kasturi Haldar

Editor-in-Chief

PLOS Pathogens

orcid.org/0000-0001-5065-158X

Michael Malim

Editor-in-Chief

PLOS Pathogens

orcid.org/0000-0002-7699-2064

Reviewer Comments (if any, and for reference):

Reviewer's Responses to Questions

**Part I - Summary**

Reviewer #1: The manuscript by Hioe et al tests an important hypothesis about the role of non/poorly neutralizing antibodies in mediating clinically relevant antiviral activities of HIV-specific antibodies in a humanized mouse model. The work is generally well and thoroughly done. The results are impactful and timely.

Reviewer #2: This manuscript analyzes the protective effect of two non-neutralizing mAbs directed to highly immunogenic epitopes V2i and V3 regions of HIV. The effect of non-neutralizing Fc-mediated inhibitory functions in HIV protection is highly suspected but not firmly demonstrated. In fact, in vivo animal models have many limitations, hampering the study of the role of FcR-functions in vivo. The humanized mouse model used in this study has the advantage to display humanized immune cells with the exact human FcR expression.

In this manuscript, the FcR function in HIV protection was extensively introduced and nicely discussed, giving a complete and comprehensive overview of the current knowledge in the field. The results of this study support previous studies, again suggesting the involvement of Fc-mediated antibody functions in in vivo decreased virus load. In this study however, authors show that Abs directed against V3, a highly immunogenic epitope, was found to have partial protective effect in vivo. Moreover, they could associate this decreased viral load the Fc domain of Ab and with in vitro ADCP function. This study is therefore relevant as it enlarges the epitopes and Fc-mediated function of Abs potentially involved in in vivo decreased viral load.

**Part II – Major Issues: Key Experiments Required for Acceptance**

Reviewer #1: None.

Reviewer #2: (No Response)

**Part III – Minor Issues: Editorial and Data Presentation Modifications**

Reviewer #1: IgG3 correlate in RV144, and association of IgG3 with elevated ADCP, may be worth specific mention in the introduction. Also may be worth mentioning HVTN 505 case-control correlates results (for example, elevated ADCP).

WNV and influence non-nAb mentions are out of place in introduction. Also unclear why those and not other viruses are mentioned.

Consider acknowledging the diversity of ADCC assays, which often show different activities for mAbs. These Abs may exhibit different relative activities in other ADCC assays.

Based on how the affinity measurements were made, they likely reflect avidity measurements (bivalent).

The authors should address the disconnect between NHP and humanized mouse models. For example, PGT121 dependence on effector function differs between models. This should be discussed as the finding of KA importance may be unique to mouse. What do the authors suppose explains the non-reduced activity of LALA, which also reduces complement activation?

Comment on biologicial relevance of readouts that were and were not statistically impacted by mAb administration.

The caveats of assessing complement activities in vitro that are relevant in vivo should be clearly acknowledged. For example, the ability of Ab to link gp120 on an ELISA plate to C1q binding does not mean that that same antibody will have any complement activity in other assays (or in vivo).

The authors should be careful to draw attention to the divergence in LALA and KA phenotypes relative to their nominal phenotypes. Ie: LALA was only an intermediate knockout of C1q binding in one of the two complement assays used.

Can the authors comment on the reproducibility of ADCP data in fig 5A? The KA titration curve appears to have no error bars and strong hook effect. Understanding the reproducibility of that dose response profile may contribute to its interpretation.

Authors should comment on statistical and biological significance, especially in light of many readouts having large spread, and some experiments having relatively few animals (ie: fig 6D) (suggesting limited resolution to meet arbitrary significance thresholds).

Why is there no experimental variance in the mAb 860 virus capture? (Fig 3B)

Reviewer #2: A few question nonetheless rises. Are the two mAbs used in this study produced by reconstitution of the Fc heavy chain domains therefore leading to the same Fc reconstituted domain for the two Abs, or have these mAbs the heavy chain as isolated from the patient? In the second case, the capacity of the Fc domain to bind to FcR may also varied considerably for these two Abs, therefore affecting FcR mediated functions. Consequently, not only the Fab domain recognizing the virus but also the Fc domain binding to FcR may influence FcR functions. This point need to be discuss especially for ADCC results.

Additional comments:

The RNA as AUC, vRNA in spleen or cell-associated RNA measured following 2219 treatment in figure 1C, D, E suggest two populations of mice: responders versus non responders. This heterogeneity may also be the results of difference of virus replication in the 3 independent experiments performed. Indeed, according to supplemental table 1, virus replication was lower in control group of experiment 63. Colors codes of the dots according to the experiment figure 1C, D, E may help to decipher this point.

To assess ADCC, authors performed 3 different in vitro assay. In order to better understand the outcome of each assay and facilitate understanding, the principle and limitations of each ADCC assay may be summarized in a table.

PLOS authors have the option to publish the peer review history of their article (what does this mean?). If published, this will include your full peer review and any attached files.

Reviewer #1: No

Reviewer #2: No

Figure Files:

Data Requirements:

Reproducibility:

References:

---

## [Decision Letter · Decision Letter 1]

9 Dec 2021

Dear Dr. Hioe,

We are pleased to inform you that your manuscript 'Non-neutralizing antibodies targeting the immunogenic regions of HIV-1 envelope reduce mucosal infection and virus burden in humanized mice' has been provisionally accepted for publication in PLOS Pathogens.

Best regards,

Daniel C. Douek

Associate Editor

PLOS Pathogens

Richard Koup

Section Editor

PLOS Pathogens

Kasturi Haldar

Editor-in-Chief

PLOS Pathogens

orcid.org/0000-0001-5065-158X

Michael Malim

Editor-in-Chief

PLOS Pathogens

orcid.org/0000-0002-7699-2064

Reviewer Comments (if any, and for reference):

Reviewer's Responses to Questions

**Part I - Summary**

Reviewer #1: The authors have made appropriate changes to the manuscript.

Reviewer #2: Authors answer my strength and queries accordingly.

**Part II – Major Issues: Key Experiments Required for Acceptance**

Reviewer #1: None.

Reviewer #2: (No Response)

**Part III – Minor Issues: Editorial and Data Presentation Modifications**

Reviewer #1: Apologies for any confusion about comments about statistical and biological significance - they were intended to encourage the authors to consider the possibility that differences that failed to meet statistical significance may reflect small sample size and the high variability observed in mouse models more than lack of biological relevance.

Reviewer #2: (No Response)

PLOS authors have the option to publish the peer review history of their article (what does this mean?). If published, this will include your full peer review and any attached files.

Reviewer #1: No

Reviewer #2: No

---

## [Editor Report · Acceptance letter]

3 Jan 2022

Dear Dr. Hioe,

We are delighted to inform you that your manuscript, "Non-neutralizing antibodies targeting the immunogenic regions of HIV-1 envelope reduce mucosal infection and virus burden in humanized mice," has been formally accepted for publication in PLOS Pathogens.

Best regards,

Kasturi Haldar

Editor-in-Chief

PLOS Pathogens

orcid.org/0000-0001-5065-158X

Michael Malim

Editor-in-Chief

PLOS Pathogens

orcid.org/0000-0002-7699-2064